# Can Emotional Intelligence Increase the Positive Psychological Capital and Life Satisfaction of Chinese University Students?

**DOI:** 10.3390/bs13070614

**Published:** 2023-07-24

**Authors:** Jingyi Xu, Myeong-Cheol Choi

**Affiliations:** Department of Business, Gachon University, Seongnam 13120, Republic of Korea; xujy201020@163.com

**Keywords:** college students’ life satisfaction, positive psychological capital, emotional intelligence, participation in cultural and artistic activities

## Abstract

The rise of artificial intelligence (AI) has led to dramatic changes in the learning environment and living conditions of college students, who face enormous psychological challenges in the ubiquitous AI environment. Modern student-management research has focused on developing mechanisms for enhancing life satisfaction, alleviating emotional anxiety, and improving self-confidence. This study aims to investigate the influence of participation in cultural and artistic activities on college students’ life satisfaction. Through a questionnaire survey and data analysis of 708 college students, this study found a significant positive relationship between participation in cultural and artistic activities and college students’ life satisfaction. Further mediation analysis showed that positive psychological capital played a mediating role between participation in cultural and artistic activities and life satisfaction. Additionally, emotional intelligence was found to play a moderating role in this relationship, and college students with higher emotional intelligence had a more significant effect on positive psychological capital in terms of participation in cultural and artistic activities. The results of this study herein provide a new understanding of life satisfaction research among college students and offer practical guidance for promoting college students’ mental health and psychological well-being. This research also demonstrates the importance of participation in cultural and artistic activities and encourages college students to be active therein to enhance their psychological capital and improve life satisfaction. Furthermore, the cultivation and enhancement of emotional intelligence is emphasized as a key factor for college students to improve their psychological well-being through cultural and artistic activities.

## 1. Introduction

In this era of rapid development of artificial intelligence (AI), which is widely used in various fields and gradually replacing certain professions and jobs, significant contributions have been made to education, management science, and other fields [1]. Although AI has human cognitive abilities [2], it remains irreplaceable in the field of emotions and feelings; therefore, humanities, culture, and arts still play an important role in the formation and understanding of emotions and feelings. However, is it possible for AI to replace human intelligence [3]? This topic remains unexplored. Consequently, this study examines the impact of participation in cultural and artistic activities (PCAA) on college students’ emotional intelligence (EI) and life satisfaction.

The rise of AI poses a challenge to the healthy development of college students’ EI, and its inability to provide the emotional support and understanding that college students need may complicate their emotion management, mishandle the negative emotions, and even affect their mental health. The Pew Research Center conducted a survey in 2018 with 979 technologists, developers, innovators, business leaders, and activists: 63% believed that using AI would benefit people, while 37% reported otherwise [4]. Before COVID-19, college students mainly studied and lived around the campus, and socialized and built relationships normally, but due to the epidemic, college students in the past three years have been engaged in a kind of isolated study and life, and their participation in cultural and artistic activities has been greatly reduced compared with before. In addition, during the period of COVID-19, college students use online lectures to study, and the wide application of intelligent robots and smart phones are changing college life, which has a certain impact on the interpersonal relationship of college students, and the communication with their real classmates or teachers has become less, leading to the phenomenon that college students are becoming increasingly more estranged and apathetic in their interpersonal interactions, which affects the establishment of college students’ emotional intelligence system. PCAA can provide college students with avenues for emotional expression and understanding, stimulate emotional empathy, and promote EI. Positive psychological capital (PPC) and self-esteem are components of EI, and they are closely related to college students’ mental health and positive emotional states. By participating in cultural and artistic activities, college students can develop PPC and enhance their self-esteem, thereby increasing their life satisfaction.

With the continuous development of science and technology, education, and culture, cultural and artistic activities are increasingly integrated into people’s daily lives. The cultural and artistic activities of college students—an important force in social development—are a significant part of campus culture construction [5]. Life satisfaction among college students, as an important element in their physical and mental health development, has received increasing attention from various educational organizations and scholars at home and abroad [6]. Previous studies have shown that the effective use of leisure time can provide personal, physical, and psychological benefits and improve interpersonal relationships, which can subsequently increase life satisfaction [7]. Notably, physical and mental health factors are important variables that influence individual life satisfaction. Good physical health increases not only participation in various activities but also life satisfaction [8]. The quality of interpersonal relationships is another variable that influences life satisfaction. Good interpersonal relationships and attachments, together with community involvement, are key factors in increasing life satisfaction [9]. Although previous studies support that PCAA is associated with personal health and life satisfaction [10], they have not included PPC and EI as research variables; There is a need to expand on the intrinsic relationship and mechanism of action between both PPC and EI.

Therefore, this study aims to investigate the effects of college students’ PCAA on their life satisfaction and the mediating and moderating variables involved thereof. Specifically, this study adopts PPC as a mediating variable, in addition to the numerous studies that have shown that EI plays a moderating role in the relationship between individual effect and emotion management. Existing studies have shown some variability in the effects of EI on college students’ life satisfaction. Therefore, this study also explores the moderating role of EI and further analyzes its underlying mechanisms.

This study can provide college students with more motivation and reasons to participate in cultural and artistic activities, provide references for the formulation and implementation of related policies, and be innovative in terms of college students’ management systems and aesthetic education. Globally, it is necessary to study the management of college students, and exploring the life satisfaction of college students can not only provide new suggestions for college education but also contribute to the academic field.

## 2. Literature Review

### 2.1. Participation in Cultural and Artistic Activities

Artistic and cultural activities are leisure exercises that improve the quality of life and also provide physical, psychological, social, educational, and aesthetic benefits, thus increasing life satisfaction [11,12].

In previous studies, cultural and artistic activities are economic and political, and they have characteristics independent from social activities, characterized as social activities that play an imperative role in the development of the individual, social unity, and growth [13]. In a narrow sense, participation in music, dance, art, theater, literature, live creative technologies, or other artistic and cultural fields recognized by competent authorities can be considered artistic and cultural participation [14]. As part of the cultural construction of college students, college students’ cultural and artistic activities have played a certain positive role in guiding their thoughts, psychology, consciousness, value identity, and behaviors [15]. College students’ PCAA can be considered a form of cultural and artistic education. Through PCAA, colleges and universities promote integrating such activities into the educational system, thereby transferring cultural and artistic knowledge and skills through education and improving the cultural literacy and competence of individuals. As an element of campus culture, cultural and artistic activities centrally reflect the common values in all students and teachers; profoundly influence students’ value orientation, ideological and moral formation, and lifestyle choices; and are a form of social culture with an educational orientation [16]. College students’ cultural and artistic activities integrate aesthetic education and artistic practice, forming a variety of types, lively and interesting ways of organizing a variety of activities, reflecting the concept of people-oriented teaching and learning, and playing a very important role in the physical and mental health of students and improvement in their comprehensive literacy [17].

Furthermore, cultural and artistic activities can also enhance college students’ sense of identity with the community, organizations, and society, thus strengthening social capital at the group level. College students can learn about history, culture, and values and experience the diversity and richness of society through PCAA. These feelings of identity and belonging can stimulate college students’ sense of social responsibility and obligation, which, in turn, can increase their life satisfaction.

### 2.2. Positive Psychological Capital

Positive psychology is a theory developed by Seligman (1998). The theory is further developed from the main field of psychology, with the goal of alleviating and treating mental illness, and focuses on the positive factors and adaptability of individuals, self-actualization, and personal growth. The subjects of positive psychology studies are divided into three parts: positive states, traits, and units [18]. Positive states refer to positive emotions, happiness, satisfaction, love, and so on, while positive traits refer to consistent behavioral elements consistently observed in individuals, including courage, wisdom, and endurance. As positive psychology focuses more on academic research and real-world interventions, it emphasizes methods of enhancing well-being and implementation [19]. Therefore, student guidance using positive psychology can actually develop students’ strengths and talents and improve their positive emotions.

Positive psychological capital is a concept developed from Seligman’s theory of positive psychology, introduced by Luthans as four measurable or developable factors: self-efficacy, hope, optimism, and resilience. These four factors are closely related to an individual’s positive cognitive state, and as a superordinate constitutive concept, PPC is widely used [20,21]. PPC refers to the positive psychological state formed through the interaction of four factors: self-efficacy, hope, optimism, and resilience. PPC can be developed and managed through personal effort and a variety of learning and training systems [22]. Therefore, if schools incorporate the concept of PPC into adolescent education, it will help to improve adolescents’ school adjustment and prevent school failure.

### 2.3. Emotional Intelligence

The theory of EI suggests an inextricable link between emotion and cognition, and that emotion can influence an individual’s thinking, behavior, and physiological responses, while cognition can influence their emotional experience and expression. Among the research on EI in the field of psychology, the theory of EI proposed by Salovey and Mayer is widely accepted. They believe that EI consists of four areas: perceiving, using, understanding, and managing emotions. This model of EI is also known as the “competence model.” EI is understood as the ability to regulate personal emotions and those of others and to effectively use what is known about emotions [23]. Research on EI in education has focused on how it affects students’ learning and behavior. Students with high levels of EI are more likely to do well in school, adapt to school and social environments, and avoid undesirable behaviors. EI is positively correlated with college students’ self-efficacy [24]. On the one hand, people with high EI have more developed social skills, are more prosocial, have less conflict, and are better at coping with emotional difficulties [25]. On the other hand, people with low levels of EI are more likely to experience interpersonal difficulties and serious psychological problems [26]. Research on EI in management has focused on how EI affects leadership and organizational performance. For example, leaders with high EI are more likely to exhibit better leadership skills at work, earn the trust and cooperation of their employees, and improve their weave performance. EI is thought to mitigate the negative effects on mental health [27]. Higher EI is associated with higher satisfaction [23], job success [28], and better health [29].

### 2.4. Life Satisfaction of College Students (LSC)

In academia, happiness, mental health, quality of life, and life satisfaction are the main indicators of an individual’s life situation [30]. Subjective well-being and life satisfaction are important determinants [31]. Happiness, mental health, quality of life, and life satisfaction have similar meanings; they usually indicate positive mental states. Park (1977) asserts that satisfaction refers to the factors that influence a person’s level of satisfaction, including both material and psychological aspects. It represents an individual’s psychological or subjective feelings, a subjective emotional state when a specific goal or need is achieved. Burr (1970) classified satisfaction into two types: (1) the degree of agreement between what an individual expects and what is actually compensated, and (2) the individual’s subjective feelings, including satisfaction and dissatisfaction, happiness and misfortune, and pleasure and unhappiness. The concept of life satisfaction has been generalized by Havighurst, Neugarten, and Tobin (1961). Neugarten (1961) and others have argued that life satisfaction is the pleasure experienced from the activities that constitute daily life. It is defined as feeling meaningful and responsible for one’s life, feeling that one has achieved one’s goals, having a positive self-image, perceiving oneself as worthy, and maintaining optimistic attitudes and emotions [32]. Life satisfaction reflects an individual’s conscious experience of inner pleasure and encourages the active pursuit of such a person’s goals [33]. Individuals with specific goals are more capable of improving their life satisfaction because they can organize and integrate their resources to achieve their goals [34]. There are several other measures, including the desire to improve life, personal satisfaction with the past and future, self-assessment of personal life, comparison of personal aspirations with actual achievements, discussion of quality of life and other indicators of physical and mental health, work and family domain aspects, and personality traits [35].

Life satisfaction is closely related to the degree of basic needs satisfaction, social support, quality of interpersonal relationships, and coping strategies. Additionally, college students’ subjective well-being is related to personality traits, psychological resilience, social support, and self-determining factors. All of these studies corroborate the theoretical basis of life satisfaction at the empirical level and provide theoretical support and practical guidance for understanding and improving college students’ life satisfaction.

## 3. Theory and Hypotheses

### 3.1. Participation in Cultural and Artistic Activities and Positive Psychological Capital

Individuals who participate in cultural and artistic activities strengthen their power of self-management, develop reflective attitudes, enhance their abilities to live with cultural sensibility, and create smooth communication [13,36]. Specifically, the four components of PPC, namely, self-confidence, resilience, hope, and optimism, are intrinsic behavioral drivers for college students. College students with high indices of PPC measures can participate more in arts and cultural activities. For example, self-confidence allows college students to overcome the difficulties and challenges they may encounter when participating in cultural and artistic activities; perseverance allows college students to persevere and revive themselves even when they encounter setbacks; hope motivates college students to keep trying and exploring new cultural and artistic activities; and optimism allows college students to face life and challenges more positively and optimistically. Some studies have shown that cultural and artistic activities can heal college students’ mental health problems. An expression referred to as art therapy is a mental health service approach that combines art and psychology, originating in the 1940s and 1950s with the study of psychiatric artists and spread rapidly worldwide [37]. Art therapy is a psychotherapeutic approach that uses art forms as a basic communicative mode to achieve the expression of the client’s inner self and thus change cognition, emotion, and behavior [38].

It is important to enhance the level of the psychological capital of college students to equip them with the ability and psychological quality to adapt to society and withstand setbacks [39]. Thus, the cultural and artistic activities of college students evidently play an important role in their mental health. PPC, which is significant for the development of mental health, can be influenced by cultural and artistic activities. According to the aforementioned theoretical basis, PCAA can enhance the PPC of college students.

As a result, the following hypothesis is proposed:

**Hypothesis** **1** **(H1).**
*Participation in cultural and artistic activities positively affects positive psychological capital.*


### 3.2. Participation in Cultural and Artistic Activities and the Life Satisfaction of College Students

Active cultural and artistic activities improve the quality of life of those who enjoy culture and serve as a channel for mutual communication in a multicultural society, and as a means of communication, individuals enable others to learn the lifestyle necessary to maintain social health [40]. Numerous studies have shown the link between leisure and happiness, leisure and fulfillment, and leisure and life satisfaction [41,42]. Cultural and artistic activities are considered leisure activities. The physical and psychological experiences, satisfaction, and interpersonal relationships involved in participation thereof promote physical, psychological, and social benefits [43]. Groves (1981) showed that employees who participated in leisure activities and engaged in their desired activities for the first six months of a two-year period increased their job satisfaction and productivity by 10% and 42%, respectively, among the less skilled group. Some studies have shown that physically, mentally, and according to the type of leisure participation, such as recreational, more leisure activities lead to greater job satisfaction [44]. Participants in leisure activities have greater job satisfaction than nonparticipants [43]. Therefore, the effect of PCAA on college students’ life satisfaction remains an important research question. Cultural and artistic activities can help people to lead more active and dynamic lives than humans can [45]. Notably, the cultural and artistic activities of college students are an important part of aesthetic education in universities, and previous studies have shown that cultural and artistic education helps the participating individuals achieve rich and harmonious human development [46]. The experience of cultural and artistic education gives a sense of satisfaction to the community residents and has a positive effect on the quality of life, and as a cultural benefit, an enriched life affects the well-being of individuals [47].

Moreover, culture and arts education can be a substantial policy tool to improve the quality of life of people, and aesthetics is an important channel for integration into society, using “beauty” as a link to connect students with society. Accordingly, colleges and universities should pay attention to arts education, introduce the needs of the times for arts and the elements of arts in the new era into the campus, create a healthy aesthetic atmosphere, make students open their aesthetic eyes, accumulate aesthetic experiences, participate in aesthetic activities, understand the core of beauty from multiple perspectives in aesthetic practice, see the charm of beautiful things, and gain spiritual power to pursue a better life [48]. PCAA is positively correlated with psychological health, and there is a causal relationship between psychological health and life satisfaction [49,50]. Therefore, college students’ PCAA can help them cultivate a broader range of interests, a richer spiritual life, and a wider social circle, along with allow them to better adapt to college life, improve their self-management skills, and enhance their lives.

Therefore, the following hypothesis is proposed:

**Hypothesis** **2** **(H2).**
*Participation in cultural and artistic activities positively affects life satisfaction of college students.*


### 3.3. Positive Psychological Capital and the Life Satisfaction of College Students

Psychological capital is the positive psychological state exhibited by an individual during growth and development [51]. In this study, PPC refers to a person’s internal resources, including self-confidence, determination, self-discipline, positive emotions, and optimistic beliefs. Previous studies have shown that psychological capital is positively associated with higher levels of life satisfaction [52]. In a previous study on corporate employees, psychological capital was significantly and positively associated with job satisfaction and psychological well-being in the relationship between psychological capital and employee well-being [53]. The concept of PPC was developed in an organizational context; however, based on research linking psychological capital to universally important outcomes, scholars have designed and validated generic measures of PPC constructs and applied them to all domains of life [54]. It has been established that psychological capital in different domains (interpersonal, health, work, and overall) is associated with positive effects, perceived social support, and life satisfaction. PPC also increases faster and creates individuals with higher life satisfaction. PPC has been more closely linked to life satisfaction [55]. These prior studies suggest a correlation between PPC and satisfaction. It has also been shown that employees with optimism in their psychological capital have higher levels of well-being than pessimists. Psychological capital is a significant predictor of both job and life satisfaction [56].

In real life, college students may face various difficulties and challenges, such as study pressure, employment difficulties, and interpersonal relationships. PCAA helps college students to improve their PPC, thus making them more optimistic, confident, and determined, and better able to cope with various challenges in their lives. In studies measuring physical activity among college students, physical self-efficacy was found to be positively related to exercise attitudes and life satisfaction among college students, and multiple regression analyses showed that physical self-efficacy significantly predicted life satisfaction [57].

Therefore, improving college students’ opportunities and motivation to participate in cultural and artistic activities can help them better adapt to and enjoy college life, while improving their quality of life and satisfaction.

Therefore, the following hypothesis is proposed:

**Hypothesis** **3** **(H3).**
*Positive psychological capital positively affects the life satisfaction of college students.*


### 3.4. The Mediating Role of Positive Psychological Capital

Research on PPC covers various fields. A previous study analyzed the mediating effect between college students’ perceptions of family health and PPC and their relationship with life satisfaction [58]. The results revealed a stable positive relationship between perceived family health, gratitude tendencies, PPC, and life satisfaction. Other studies have analyzed the structural relationships among PPC, basic psychological needs satisfaction, autonomy, sense of competence, interpersonal relationships, academic engagement, and life satisfaction among college students [59]. The results of this study show that PPC of college students directly affects their academic engagement and life satisfaction. In a study on the association between leisure preference, positive thinking, psychological capital, and life satisfaction, PPC was found to mediate the relationship between leisure preference and life satisfaction as well as the link between spending time with family, participation in activities, and life satisfaction [60]. Other studies have analyzed the relationship between high school students’ leisure activities and life satisfaction through structural equation modeling and examined the mediating effect of perseverance and PPC [61]. It was shown that high school students’ leisure activities predicted their life satisfaction and that perseverance and PPC mediated the relationship between leisure activities and life satisfaction. This study also verified that the dual mediating effect of perseverance and PPC in the relationship between leisure activities and life satisfaction was significant.

Based on the results of previous studies, it can be inferred that better outcomes can be expected when the perception of PPC is higher. Thus, these studies support the application of PPC across multiple disciplines and simultaneously demonstrate [62] that greater synergistic effects can be achieved when PPC, as a comprehensive concept, is combined with other constructs [63]. In addition, based on the results from various aspects of prior research, it can be inferred that an increase in the level of PPC among college students may positively affect cultural and artistic activities; therefore, it is desirable to investigate the relationship between PPC and college students’ life satisfaction. For example, college students’ PCAA may enhance their optimism and sense of hope, and these positive emotions and mindsets may reduce their anxiety and depression, thereby increasing life satisfaction.

Therefore, the following hypothesis is proposed:

**Hypothesis** **4** **(H4).**
*Positive psychological capital plays a mediating role between cultural and artistic activities and life satisfaction of college students.*


### 3.5. Moderating Effect of Emotional Intelligence

The social benefits of cultural and artistic activities and EI positively influence each other in a close relationship, and through the social effects of cultural and artistic activities, self-confidence increases, while concern for regional interactions and social power is strengthened, albeit as a result of an increased ability to recognize each other’s moods and regulate sensuality [64]. Studies on the effects of fun engendered by cultural and artistic activities have shown that, for adolescents, there is a positive correlation between music in reducing stress and restoring sensuality [65]. In addition, for the development of EI and three-dimensional arts program activities for children, the results of input activities for producing work, pleasure, creativity, and self-esteem are on the rise, as are their feelings. There appears to be an adjustment, and active expression of sensual results; thus, for careful analysis, it is necessary to increase the time and number of artistic activity inputs [66]. There are also studies on the social effects of cultural and artistic activities on the development of EI that conclude that dance activities contribute to the development of EI by making adolescents positively aware of their bodies, thus acquiring a rich and varied sensual expression that can effectively regulate their own and others’ sensuality, and so on [67]. Studies have shown that social effects such as happiness, cooperative spirit, and creativity are generated through after-school dance activities, resulting in positive changes in students’ sensuality and the emergence of self-regulation [68].

In summary, cultural and artistic activities affect the development of the required EI, which has been confirmed in many prior studies. Studies on adolescent participation in experiential activities have found that, for both male and female adolescents, PCAA through experience has the same effect on the EI of students. Social and emotional intelligence reintroduces positive perception changes as a result of a static impact on life goals and knowing how to enjoy oneself, live creatively, and take risks [69]. An increase in EI can help college students reduce stress, resolve conflicts, build and maintain relationships, and improve their social lives.

Therefore, the following hypothesis is proposed:

**Hypothesis** **5** **(H5).**
*Emotional intelligence has a moderating effect on college students’ participation in cultural and artistic activities and their life satisfaction.*


However, little research has been conducted on EI and PPC [70]. Earlier studies assessed the development of PPC and EI separately but neglected the effectiveness of educational interventions targeting both [71]. Nathalia and Lisete emphasized the importance of the interrelatedness of EI and PPC in achieving organizational outcomes and goals, and increasing productivity. The results showed a significant relationship between EI and PPC. The results showed that employees’ emotional control and motivation were most significantly correlated with PPC. A study on EI and PPC in Korea showed that the higher the EI, the higher the self-efficacy and optimism as psychological capital [72], with the result that people with a good self-understanding of feelings can regulate them, use them, and thus perceive and understand others’ feelings and have higher self-efficacy and stronger optimism than those who cannot. In this study, the subjects were university students, and for this particular group, it was argued that EI moderated the relationship between cultural and artistic engagement and PPC.

Therefore, the following hypothesis is proposed:

**Hypothesis** **6** **(H6).**
*Emotional intelligence has a moderating effect on participation in cultural and artistic activities and positive psychological capital.*


The above hypotheses can be summarized as shown in Figure 1, which illustrates the research model.

## 4. Methods

This study used SPSS 26.0 and bootstrap for data analysis. AMOS 24.0 was used for confirmatory factor analysis (CFA) and structural equation modeling (SEM).

### 4.1. Measures

The Culture and Art Activity Participation Scale was obtained from the Ministry of Culture, Sports, and Tourism’s (Korea, 2014) National Leisure Activity Survey. Cultural and art viewing activities include viewing actions to connote cultural and art performances, such as visual viewing of exhibitions, museums, music recitals, theater performances, dance performances, acting performances, and so on. Cultural and art participation activities involve direct participation in cultural and art performances such as literary activities, literary creation/reading discussions, fine arts activities, instrument playing/singing classrooms, traditional arts, photography, dance, and other activities directly involved in cultural and artistic performances, creative activities, fine arts, and performances. This study combined the two dimensions to form the Cultural Arts Activity Participation Scale, with 16 questions in total.

The Positive Psychological Capital Scale used the scale developed by Luthans et al. (2007), adapting the questions to the college student population. The Positive Psychological Capital Scale developed by Luthans et al. is mostly related to job and goal attainment in organizations, thus improving the limited research framework with college students and the general population, which has been tested by many scholars for its reliability. The scale consists of 4 dimensions with 10 question items.

The scale of EI used was developed by Wong and Law [23] and this scale is widely used in the academic field and has good reliability. The scale consists of four dimensions, namely, self-emotional understanding, emotional understanding of others, emotional application, and emotional regulation, with eight questions.

The Life Satisfaction Scale was selected from “Structure and Scale Development of Life Satisfaction of Adolescent Students” [73], edited by Zhang et al. (2004). The Satisfaction with Life Scale [74,75] has been used extensively as a measure of life satisfaction component of subjective well-being since its introduction in 1985. Eight question items were used for this study.

### 4.2. Participants and Procedure

In this study, college students enrolled in eight universities were selected, with Shanxi Province and Guangxi Province as the main regions. The majors involved were humanities and social sciences, science and technology, arts and physical education, and medicine. The questionnaires were distributed on April through May 2023 by the teachers and counselors employed at each school. The online questionnaires were collected, and the students were assured of confidentiality. A total of 756 questionnaires were returned, excluding unqualified questionnaires, and 708 questionnaires were used for this study, representing a questionnaire recovery rate of 93.65%.

### 4.3. Control Variables

The gender, age, educational background, and job position of the surveyed participants have been controlled for in many studies. In this study, limited by the type of survey participants, the gender, grade, and profession of the surveyed participants were controlled for, and this information could influence the relationship between the variables, where gender was coded as a dummy variable: 1 for male and 2 for female.

## 5. Results

### 5.1. Descriptive Analysis

Based on the data collected from 708 participants, the descriptive line statistical analysis of the demographic variables showed that 51.69% of the participants were males. The percentage of females was 48.31%. More than 50% of the participants were in their “first year” of undergraduate studies. Further, 32.77% were in the second year, with a higher percentage of these two grades (approximately 90%). In total, 41.81% of the majors were ”Arts and Sports”. The percentage of participants studying science and engineering was 40.68%. The percentages of these two majors are the highest. The specific data are presented in Table 1.

### 5.2. Model Validation Test

In this study, the data for each variable were analyzed via descriptive statistics and Cronbach’s α test through the SPSS 26 software. From Table 2, it is evident that the mean value for PCAA is 1.928 and the median value is 1.75, both of which are less than 3, indicating that the participation of college students in artistic activities, in general, is very low. In contrast, the mean and median of several variables of PPC, EI, and life satisfaction are above 3, indicating that the PPC, EI, and life satisfaction of the surveyed participants are at a high level. According to the results of the Cronbach’s α index in the above-presented table, we can find that the Cronbach’s α values for each variable are above 0.7; the reliability of this survey is excellent.

This study used validation factor analysis to test the convergent and discriminant validity of the questionnaire. In this study, validatory factor analysis (CFA) was conducted using AMOS 24.0 software. The results of the model fit follow: χ2/df = 3.037, less than 5; CFI = 0.941, NFI = 0.915, and IFI = 0.942, all greater than 0.9; SRMR = 0.047, less than 0.05, and RMSEA = 0.054, less than 0.08. Overall, this indicates that the data duly fit the validated factor analysis model.

In this study, standardized factor loadings, combined reliability, mean variance extraction, and the arithmetic square roots were used to measure the convergent validity of the sample. As shown in Table 3, the average variance extracted (AVE) values for each variable was greater than 0.5 and the composite reliability (CR) values were greater than 0.7. This indicates that the convergent validity of the questionnaire is qualified. The square root of the AVE value was taken and compared with the correlation coefficients of the variables and others, as shown in the table above. The number on the diagonal (AVE value) is greater than any of the correlation coefficients in its column; therefore, the discriminant validity test is passed. Thus, the questionnaire passed the validity test.

In this study, the variables were tested for common method bias using one-way confirmatory factor analysis (CFA). Validating factor analysis was performed on all data using AMOS 24.0 software, and the results of the one-way model were not acceptable. χ2 = 18,812.956 was much larger than the study model of 3110.366, χ2/DF = 18.194 was much larger than 5, CFI = 0.501 was much smaller than 0.9, NFI = 0.488 was much smaller than 0.9, IFI = 0.502 was much smaller than 0.9, SRMR = 0.176 was much greater than 0.1, and RMSEA = 0.156 was much greater than 0.1. This indicates that the data in this study do not suffer from common method bias.

### 5.3. Hypothesis Testing

To demonstrate whether the variables correlated with each other, a Pearson correlation analysis was performed on each variable by using SPSS 26. The results are shown in Table 4.

The correlation coefficients between LSC and PCAA, between LSC and PPC were significant at 0.668 and 0.531, respectively, all of which were greater than 0. This indicates a positive relationship between LSC and PCAA, between LSC and PPC.

The correlation coefficient between PPC and PCAA was 0.611 with a 0.01 level of significance, thus indicating a significant positive relationship between PPC and PCAA. Therefore, Hypothesis 1 is supported.

In this study, stratified regression analysis was performed to test the stability of the model. As shown in Table 5, two models were used in the hierarchical regression analysis. The independent variables in Model 1 were gender, grade, and major; in Model 2, PCAA and PPC were added to Model 1. The dependent variable of the model was LSC. As can be seen in Table 5, the regression coefficient value for PCAA is 0.447 and shows significance (t = 13.667, *p* = 0.000 < 0.01), implying that PCAA has a significant positive effect relationship on LSC. This shows that Hypothesis 2 is supported. The regression coefficient value for PPC is 0.135 and shows significance (t = 3.472, *p* = 0.001 < 0.01), implying that PPC has a significant positive influence relationship on LSC. This shows that Hypothesis 3 is supported.

The upper and lower limits of the bootstrap 95% interval for the mediated path PCAA => PPC => LSC total effect, indirect effect, and direct effect do not contain 0. This also indicates that PPC plays a mediating role between PCAA and LSC, and it has a partial mediation effect. The specific results are summarized in Table 6.

The mediation path PCAA => PPC => LSC bootstrap test shows that the total and direct effects are less than 0.001. The indirect effect value for PPC in PCAA and LSC is 0.1013 (LLCI = 0.0252, ULCI = 0.1826, interval exclusion 0) accounted for the total effect value. This also indicates that PPC plays a mediating role between PCAA and LSC, and it has a partial mediation effect. This indicates that PCAA can not only directly affect LSC but also affect LSC through PPC. Therefore, Hypothesis 4 is supported by this finding.

From Table 7, the Model 1 data show that the interaction term product coefficient β of EI and PCAA is 0.044, with a confidence interval (LLCI = −0.004, ULCI = 0.092) containing 0. This proves that the moderating effect of EI on the relationship between PCAA and LSC is not significant. Thus, Hypothesis 5 is not supported. Data from Model 2 prove that EI moderates the relationship between PCAA and PPC, thereby implying that the moderating variable (EI) has a significant difference in the magnitude of the effect at different levels concerning the effect of PCAA participation on PPC, as shown in the simple slope plot (Figure 2).

As can be seen from Figure 2, the linear relationship between participation in PCAA and PPC is flatter when EI is at low levels. The slope of the linear relationship between PCAA participation and PPC is larger when EI is at a high level than at a low level. This indicates that this moderating effect is positive, and thus, Hypothesis 6 is supported.

## 6. Conclusions

### 6.1. Results and Discussion

This study analyzed the effect of PCAA on college students’ life satisfaction, and the mediating effect of PPC and the moderating effect of EI. The results of this study are as follows:

First, PCAA has a significant positive effect on college students’ life satisfaction. PCAA can provide rich emotional experiences and psychological fulfillment, making college students feel more fulfilled, meaningful, and satisfied with their lives. This study justifies the significant impact of arts and cultural participation on an individual’s quality of life or any other indicator of physical health and well-being [76,77,78]. Arts and cultural participation is not only an important component of the public’s leisure participation behavior but also an effective means of enhancing personal life satisfaction [79]. Previous studies have reported a positive relationship between leisure participation and life satisfaction [80]. In particular, greater participation in arts and cultural activities is associated with increased life satisfaction [81]. This study was validated in a college student population, whereby higher arts and cultural participation enhanced the overall life satisfaction of college students.

Second, PPC plays a mediating role between PCAA and college students’ life satisfaction. Participation in arts and cultural activities fosters PPC among college students, including self-confidence, optimism, and resilience, which in turn enhances their life satisfaction. AI has a negative impact on college students’ values. The network environment is becoming more complex, negative information is more hidden; some antimainstream values or unhealthy ideas will have a greater impact on college students, and some students who are weak in distinguishing between right and wrong will even pass some false, vulgar, or even reactionary remarks, which is not conducive to the establishment of a positive psychological state of college students. And with the popularization of artificial intelligence application, negative information will make the value orientation of college students deviate, artificial intelligence technology causes some college students to encounter frustration in the process of study or job hunting, which makes college students reduce self-efficacy and increase negative emotions [82]. The more people engage in positive behaviors, the less likely they are to experience negative emotions [83]. Individuals can derive psychological benefits from leisure participation [30], which correlates with several empirical findings. Life satisfaction is an important conceptualization of positive psychology and an important indicator of subjective well-being [84]. A previous study showed that psychological capital was positively associated with higher levels of life satisfaction [52]. Individuals who scored higher on the Positive Psychology Questionnaire items were more likely to engage in opportunities to maintain and improve their well-being and to persist in achieving their goals [85]. In a study on the relationship between life satisfaction and psychological capital among Chinese students, the results showed that students’ satisfaction with life was directly predicted by PPC itself [86]. Many studies have shown an association between PPC and life satisfaction or PPC and quality of life [87,88]. Recently, the *Survey Report on Network Literacy of Digital Youth in the New Era* (2023) released by Beijing Normal University, shows that the overall average score for network literacy of college students is 3.67 (out of 5), and the score for teenagers and junior high school students is 3.56, which is slightly higher than the passing line, and is in urgent need of improvement. In the era of information explosion and rapid iteration of intelligent technology and the rapid development of artificial intelligence technology, the growth of students is an issue that many experts in the field of education have called for social attention [89]. It is thus clear that the psychological condition of college students in the era of AI requires more attention. This study, after analyzing the statistical results, confirms previous studies and extends the field to the college student population, where the mediating role of the PPC of college students is significant and enhances the quality of life of college students while they are physically and mentally satisfied by actively participating in cultural and artistic activities.

In addition, EI plays a moderating role between PCAA and the psychologically positive capital of college students. In a previous study on EI, resilience, and self-esteem as predictors of life satisfaction among college students, the results indicated that EI had an influential role on life satisfaction [90], and the results from analyzing Hypothesis 6 indicated that college students with high EI were more capable of fully utilizing emotional expression and understanding in cultural and artistic activities, wherefrom they derived emotional support and emotional wisdom; however, Hypothesis 5 was rejected, suggesting that EI had a moderating role in this context. This indicates that there is no moderating effect of EI on the relationship between PCAA and life satisfaction of college students for several reasons. First, the scale chosen is a mature scale widely used internationally, and although it has high reliability, it is not fully applicable to Chinese college students, and certain semantic bias exists. Second, in the process of sampling, the scope of selection was narrow, mainly focusing on colleges and universities in Shanxi and Guangxi provinces, and the questionnaire was not collected nationwide. This may have led to some limitations in the participant data. Future research can expand the sample size and scope to global dimensions and refine the type of sample to reduce errors in the research results. Third, according to the theoretical basis of this study, the process of choosing cultural and artistic participation by college students is more subjective, EI is mainly reflected in interpersonal communication and life, and high or low EI cannot directly affect the choice of cultural and artistic activities, while this selection process is complicated for college students themselves. Therefore, this hypothesis was not supported. The findings of this study have important theoretical and practical implications for college students’ education and mental health support.

### 6.2. Theoretical and Practical Implications

From a theoretical perspective, this study expands our understanding of the mechanisms that shape college students’ life satisfaction by exploring the relationships among PCAA, PPC, and EI. These findings support the mediating role of PPC between PCAA and life satisfaction along with the moderating role of EI thereby. This provides new theoretical perspectives for research in the fields of psychology and education and enriches research on EI and well-being among college students.

From a practical perspective, the results of this study are instructive for the management and mental health support of college students. Colleges and universities can enhance college students’ life satisfaction by increasing opportunities and resources for cultural and artistic activities and by providing a platform for students to participate in such activities, thereby promoting the development of their EI and PPC. The findings of this study also highlight the importance of participation in cultural and arts activities for college students’ life satisfaction and suggest corresponding management and support strategies. This has important implications for improving college students’ mental health and well-being, together with value for the practice of university education and mental health support.

### 6.3. Limitations and Future Research

However, there are several directions for future research. With the rapid development of AI technology, is it possible to practically apply AI technology in the construction of college students’ cultural and artistic activities, and carry out activities that are beneficial to the development of college students’ mental health. In circumventing the disadvantages brought about by AI, the practical use of it will be expanded to all aspects of college students’ learning and life. Secondly, different cultural contexts can be taken into account in the field of research on life satisfaction of university students, as this current study mainly focuses on college student groups from specific cultural backgrounds. Future research can consider cross-cultural comparisons to explore whether differences exist in the effects of cultural and artistic activities on life satisfaction across different cultural backgrounds. Second, future studies should conduct long-term tracking surveys to understand the long-term effects of PCAA on life satisfaction and explore the changes and development trends among them. In future research, other mediating and moderating variables are considered to understand the mechanism influencing cultural and artistic activities on life satisfaction.

Furthermore, this study has some limitations. In terms of the research method, this study used questionnaires to collect data, and this method is influenced to some extent by individual subjective evaluation and memory bias. Future research can use various methods, such as field observations and in-depth interviews, to obtain more comprehensive and accurate data. The representativeness of the sample could expand the sample size and include students from different universities and regions to increase the generalizability of findings in future studies. In addition, this study lacks longitudinal data and is based on cross-sectional data analysis, which cannot observe the longitudinal effects of participation in cultural and arts activities on college students’ life satisfaction. Future research will further promote the advancement of college students’ management and mental health support to provide better life experiences and development opportunities for the students.

## Figures and Tables

**Figure 1 behavsci-13-00614-f001:**
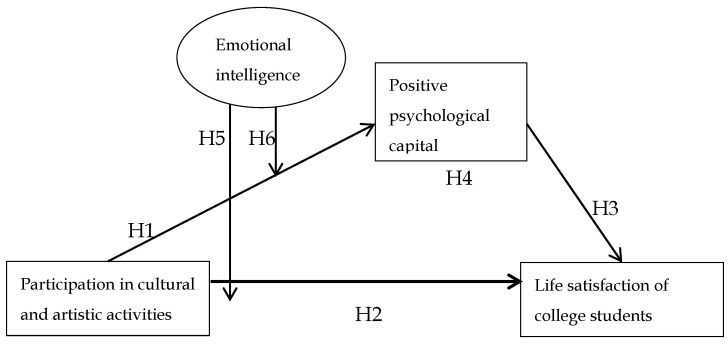
Research model.

**Figure 2 behavsci-13-00614-f002:**
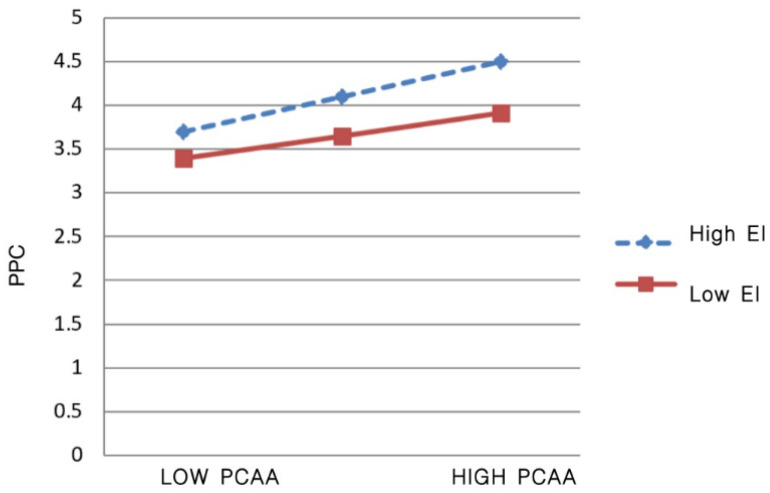
The moderating role of emotional intelligence.

**Table 1 behavsci-13-00614-t001:** Descriptive analysis of participants.

Demographic Variable	Type	Frequency	Ratio (%)
Gender	Male	366	51.69
	Female	342	48.31
Grade	First Year	396	55.93
	Second Year	232	32.77
	Third Year	43	6.07
	Fourth Year	33	4.66
	Graduate and above	4	0.56
Major	Humanities and Society	121	17.09
	Science and Engineering	288	40.68
	Arts and Sports	296	41.81
	Medicine	3	0.42
Total		708	100.0

**Table 2 behavsci-13-00614-t002:** Participant status analysis and reliability test.

	Mean	SD	Median	Cronbach’s α of Variables
PCAA	1.928	0.813	1.75	0.868
PPC	3.807	0.712	3.9	0.854
EI	3.468	0.929	3.63	0.864
LSC	3.781	0.724	3.88	0.847

Note: PCAA: participation in cultural and artistic activities; PPC: positive psychological capital; EI: emotional intelligence; LSC: life satisfaction of college students.

**Table 3 behavsci-13-00614-t003:** Convergent and discriminant validity.

	AVE	CR	PCAA	PPC	EI	LSC
PCAA	0.678	0.971	0.824			
PPC	0.689	0.956	0.611	0.830		
EI	0.772	0.964	−0.059	0.288	0.879	
LSC	0.698	0.949	0.668	0.531	0.002	0.835

Note: The number on the diagonal is the average variance extracted (AVE) value. PCAA: participation in cultural and artistic activities; PPC: positive psychological capital; EI: emotional intelligence; LSC: life satisfaction of college students; CR: composite reliability.

**Table 4 behavsci-13-00614-t004:** Correlation analysis.

	Gender	Grade	Major	PACC	PPC	EI	LSC
Gender	1						
Grade	0.057	1					
Major	−0.225 **	−0.351 **	1				
PCAA	0.011	−0.072	0.153 **	1			
PPC	0.114 **	−0.083 *	0.112 **	0.611 **	1		
EI	0.121 **	−0.033	−0.013	−0.059	0.288 **	1	
LSC	0.05	−0.071	0.138 **	0.668 **	0.531 **	0.002	1

Note: The lower triangle is the Pearson correlation coefficient between variables; * *p* < 0.05, ** *p* < 0.01. PCAA: participation in cultural and artistic activities; PPC: positive psychological capital; EI: emotional intelligence; LSC: life satisfaction of college students; CR: composite reliability.

**Table 5 behavsci-13-00614-t005:** Stratified regression analysis.

	PCAA	PPC	R²	R² (Adjustment)	F	ΔR²	ΔF
Stratification 1			0.027	0.022	F (3.704) = 6.413, *p* = 0.000	0.027	F (3.704) = 6.413, *p* = 0.000
Stratification 2	0.447 ** (13.667)	0.135 ** (3.472)	0.481	0.477	F (6.701) = 108.374, *p* = 0.000	0.455	F (3.701) = 204.768, *p* = 0.000

Note: ** *p* < 0.01; The value in parentheses is t-value. PCAA: participation in cultural and artistic activities; PPC: positive psychological capital.

**Table 6 behavsci-13-00614-t006:** Mediating role of positive psychological capital.

Independent Variables	Dependent Variable	Type of Influence	Effect	BootSE	BootLLCI	BootULCI	Effect Ratio
PCAA	LSC	Indirect Effect	0.1013	0.0399	0.0252	0.1826	17.26%
		Direct Effect	0.486 **	0.0528	0.4251	0.6341	
		Total Effect	0.587 **	0.049	0.500	0.690	

Note: ** *p* < 0.001.

**Table 7 behavsci-13-00614-t007:** Moderating effect of emotional intelligence.

Dependent Variable	PPC			LSC		
	Β	LLCI	ULCI	β	LLCI	ULCI
PCAA × EI	0.155 ***	0.117	0.193	−0.032	−0.073	0.01
R2	0.529			0.453		
F	131.034			96.889		

Note: *** *p* < 0.001. PPC: positive psychological capital; LSC: life satisfaction of college students.

## Data Availability

Data is unavailable due to privacy or ethical restrictions.

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
