# Peer review of "Can Emotional Intelligence Increase the Positive Psychological Capital and Life Satisfaction of Chinese University Students?"

_behavsci, 2023, doi:10.3390/bs13070614_

Round 1

Reviewer 1 Report

Review: can emotional intelligence increase the positive psychological capital and life satisfaction of Chinese University students?

This study considers the impact of PCAA on EI and life satisfaction. It mentions early on a hot topic in the literature now: the influence of AI in the creative industries and the potential for its penetration there. As the authors, say this intrusion into many areas of our lives has serious implications for people. I was disappointed that this idea didn’t seem to be picked up later in the paper.

Introduction

this section gives the background well, and also includes some comments about people’s general perceptions about the positive and negative impacts of AI. PPC is a strong concept within EI. It certainly seems sensible that cultural activities can increase PPC and satisfaction, as well as mental health.

Literature review

this section covers the various chunks of relevant literature citing sufficient studies, although I thought the section on PCAA was relatively short considering its centrality to the argument

Theory and Hypotheses

the hypotheses are straight-forward and theory well argued. Figure 1 is helpful to the reader

Methods

the measured used are described well. There are over 700 participant university students in the study.

Results

this section seems okay. At line 418 I’m not sure why male is in parentheses. Suitable reliability and validity measures are given for the scales utilised.  Hypothesis testing reveals that there is a positive relationship between LSC and PCAA. Just a small thing, you refer to hypothesis 3-6 specifically and perhaps you could also do this for hypotheses one and two. Most of the hypotheses are supported.

Conclusion.

I found it odd to read discussion of the results in the conclusion section. I was actually looking for greater discussion of the results close to when the results are presented. It is common practice I think to have results and discussion grouped together. Having said that, the results are compensability discussed on page 13 and 14. EI did not moderate between PCAA and LSC. There is suitable discussion about this. Discussion of theoretical and practical contributions is quite okay and presents the implications sensibly.

I think there should be a conclusion section now before the future research section.

Future research/limitations

This section seems okay, but once again given the mention of artificial intelligence upfront I think it would be quite good to introduce some comments about that here.

Overall though quite a creditable paper. I did notice a few spacing issues.

Author Response

Artificial Intelligence related research is added in the EI and Conclusion sections.

The literature review section was supplemented with studies related to PCAA, especially those in the Chinese academic field targeting PCAA in colleges and universities.

It was a mistake, it's been corrected.

Hypotheses 1, 2 are supported as reflected in the regression analysis section of the article.

Reflecting the review opinions, the structure of the conclusion was revised and the content was supplemented.

If there are any deficiencies, I will make further corrections based on your comments.

We introduced and increased the explanation and contents of artificial intelligence.

Reviewer 2 Report

Thank you very much for this opportunity to revise the manuscript titled "Can Emotional Intelligence Increase the Positive Psychological Capital and Life Satisfaction of Chinese University Students?" that was submitted to Behavioral Sciences.

I want to emphasize that the authors have done a great job on the article, and the data obtained during the study certainly have a high scientific and practical value.

The urgency and timeliness of the research, realized by the authors of the reviewed article, are obvious.

I made a few comments with the hope of assisting the authors to improve their paper.

1. It is interesting to compare post-pandemic with pre-pandemic. Are there previous studies that investigated the influence of participation in cultural and artistic activities on college students' life satisfaction before the COVID-19 pandemic? Please discussed.

2. I suggest that the authors focus their efforts on broadening their interpretations and recommendations in the Discussion section in order to improve the overall quality of the paper.

3. I suggest that the authors have their paper reviewed by a native English-speaking colleague or utilize a paid editing service.

I will be happy to review the revised manuscript.

I suggest that the authors have their paper reviewed by a native English-speaking colleague or utilize a paid editing service.

Author Response

COVID-19 has a great impact on the academic life of Chinese college students, which is a good suggestion. We will include this part in the introduction section. In addition, there are studies in the academic field on the impact of cultural and artistic activities on life satisfaction, which have been heavily cited in the article. In addition, this acting is a study conducted after COVID-19, and the data and research content are brand new.

Some changes were made in the discussion section and the future research section. A discussion of AI was added, as well as the lack of research on different cultural contexts.

Before submitting the thesis, it was supervised by a native English-speaking professor at a translation company.

In the next revision of this thesis, I will submit the thesis again with editing service.
